# Study Regarding the Optimal Dimension of Intraoral Bitewing Radiographs in Patients with Primary Dentition

**DOI:** 10.3390/ijerph192215413

**Published:** 2022-11-21

**Authors:** Montserrat Diéguez-Pérez, Concepción Lacalle-Muñoz de Cuerva, Fernando Costa-Ferrer, Marta Muñoz-Corcuera

**Affiliations:** 1Preclinical Dentistry Department, Faculty of Biomedicine and Health Sciences, European University of Madrid, 28670 Villaviciosa de Odón, Spain; 2Doctoral School and Research, Faculty of Biomedicine and Health Sciences, European University of Madrid, 28670 Villaviciosa de Odón, Spain; 3Departamento de Odontología Pre-Clínica, Facultad de Ciencias Biomédicas y de la Salud, European University of Madrid, 28670 Madrid, Spain; 4Clinical Dentistry Department, Faculty of Biomedicine and Health Sciences, European University of Madrid, 28670 Villaviciosa de Odón, Spain

**Keywords:** dental radiography, bitewing, pediatric patients, primary dentition, dimension, discomfort, rejection

## Abstract

The aim of this study was to determine the optimum dimensions of a radiographic plate to allow correct visualization of dental tissues and correct fit in the oral cavity of children with deciduous dentition. A quasi-experimental clinical study was carried out in children of both sexes aged between 3 and 5 years. The study variables were the complete visualization of the dental structures, the surveillance of ischemia on soft tissues, stimulation of the gag reflex, and acceptance of the radiographic plate by the pediatric patient through a validated visual analogue scale that measures anxiety. The data obtained were subjected to a descriptive and comparative statistical analysis carried out for both study phases. A total of 80 children participated in the study. The optimal dimensions obtained for the radiographic plate were 19.5 mm in height and 27.3 mm in width. Visualization of the dental tissues during both phases was not statistically significant (*p* = 0.412). However, there were statistically significant differences regarding the presence of ischemia, gag reflex, and child rejection (*p* < 0.001). A smaller radiographic plate allows correct visualization of the coronal dental tissues without causing rejection, ischemia, or gag reflex in patients in the deciduous dentition.

## 1. Introduction

The interproximal or bitewing technique is useful for orderly study and allows the visualization of interproximal and occlusal caries lesions. Clinical examination alone underestimates the presence of interproximal caries; in fact, according to some research, almost twice as many carious surfaces are diagnosed if bitewing radiographs are taken [1]. In addition to these lesions, pulp chamber alterations, overflowing restorations, recurrence of caries under fillings, quality of fit and finish of preformed crowns, the characteristics of the alveolar ridge, and the amelocemental boundary, among other alterations, can be visualized with bitewing radiographs. In the same film, the coronal, cervical, and furcation regions of the deciduous teeth of both arches can be observed at the same time [2,3,4] It is, therefore, an essential tool for radiographic diagnosis in pediatric dentistry but it constitutes a challenge for the practitioner. The doses of the radiographs taken and the diagnosis obtained from them should be indicated in the patient’s medical record, as well as the informed consent of the patient’s parents [4].

According to the European Academy of Pediatric Dentistry (EAPD), the principles for protecting the child from radiation in pediatric dentistry rely on its the justification for such a practice and also take into consideration the cooperation of the pediatric patient [4]. Radiography with very-low-dose exposure but without image quality is not justified at this age [4]. An error commonly performed by pediatric dentists after taking a radiographic record using this technique is image superimposition [5]. Therefore, in order to try to remedy this problem, it is important that the radiographic examination of the pediatric patient involves their cooperation; thus, unpleasant sensations should be avoided. The use of conventional radiographic plates for the intraoral bitewing technique is becoming more frequent in the pediatric dentistry field; however, it is uncomfortable and could trigger nausea, thus being rejected by many of these patients [6,7,8,9]. Altogether, it can be deduced that the dimensions of the oral cavity of patients in the primary dentition do not harmonize with the lengths of the radiographic plates currently available on the market, as these are oversized.

It was hypothesized that the use of a smaller plate size than the one usually marketed would allow pediatric patients to feel less reluctance toward radiographic examination, thus achieving a better diagnosis of interproximal caries and allowing the visualization of all relevant anatomical structures.

On this basis, we set out to determine the appropriate dimensions of the radiographic plate that allows the correct visualization of dental tissues, optimal fit in the oral cavity, and acceptance by children in primary dentition.

## 2. Materials and Methods

A quasi-experimental, clinical, observational, cross-sectional, and analytical study was designed. It was carried out in accordance with the regulations of the Research Ethics Committee of the European University of Madrid (internal code CIPI/036/17) and received the approval of the Ethics Committee of the Community of Madrid (project code 28008088).

### 2.1. Study Population, Selection Criteria, and Sample Size

Fieldwork was conducted from January 2018 to October 2019 in the Community of Madrid. The parents and/or caregivers of all child participants in the study requested a first visit or check-up at the University’s Dental Clinic for clinical and radiographic diagnosis of their children’s oral health status. The sample selection criteria were as follows: children aged 3 to 5 years with complete primary dentition stage and the absence of frank carious lesions, fillings, alterations in dental development, and other manifestations that do not allow the correct visualization of dental tissues at the coronal and furcation level (1), good general health (2), children with a percentile in the range of 3–97 (3), the need to take radiographic records with the flap radiographic technique outside the research (4), signature of the informed consent by parents or legal guardians (5), and cooperative patients (6). Patients whose caregivers refused to participate in the study (1), syndromic patients (2), and patients with pathologies affecting the correct development and growth of the child (3) were excluded.

The sample size calculation took into account the fact that not every child would want to have radiographic plate placed inside their mouth after the use of a standard radiographic plate (expected to be 90%) and a cropped plate (expected to be less than 50%) on a previous occasion. The inclusion of 80 children allowed these differences to be detected with a statistical power of more than 90% and a confidence level of 95%.

### 2.2. Data Collection, Research Systematics, and Study Variables

Two phases were established, perfectly differentiated in time (phase I vs. phase II). The previously trained and calibrated main investigator was responsible for the research systematics detailed below. At the beginning of the study, after receiving the patients, the parents and guardians were informed of the characteristics of the study and signed the informed consent form. All children were seen during the morning from 8:30 a.m. to 1:30 p.m. For patient selection, the medical and dental clinical history was reviewed. Once the winter coats were removed, height and weight were recorded with an approved scale model 7691321994 from SECA GmbH & Co. (Hamburg, Germany). KG. This was after a brief explanation of the procedure to the children taking into consideration their age. A total of five patients per shift were scanned for final selection. Only ambient light was used for both oral examination and radiographic recording. The latter was carried out using the X-mind X-ray machine with the following specifications: 70 KVP 8 mA, total filtration 2.3 mm Al/70 kV, nominal line voltage 230 V ± 50/60 Hz and a Toshiba radiographic tube.

#### 2.2.1. Phase I: Determining Tooth Dimensions by Radiographic Analysis after the Bitewing Technique

Each participant underwent a radiographic examination with DURR^®^ Brand’s (Bietigheim-Bissingen, Germany) 2 × 3 cm phosphor plates size 0+. They were previously trained to retain the plate in the mouth. The presence of ischemia and gag reflexes was inspected. At all times, the child remained seated in the dental chair with the approved plumbed apron, sized according to their anthropometric characteristics. The occlusal plane was kept parallel to the floor. To ensure that the child did not move and remained in the proper position, the main investigator remained with the leaded apron and gloves inside the box during the shot. The validated visual analogue scale used to measure anxiety was taken from Abu-Saad, showing a happy and a sad face that the child must select after the experience [10]. It was employed to measure rejection or affinity. The disposable sleeve was then removed and inserted into the Vistascan plate reader for subsequent analysis of tooth dimensions. The optimal radiographic plate size was calculated by taking as reference the most protruding point of the mesial side of the four primary canines, the most convex point of the distal side of the four primary second molars, and the most concave point of the furcation of the four primary second molars. From these anatomical points, the horizontal and vertical dimensions of the teeth from each hemiarch were determined, and 1 mm was added to the dimension obtained in the vertical and horizontal directions (Figure 1).

#### 2.2.2. Phase II: Determining the Suitability of the Trimmed Plate

Initially, the radiographic plates were trimmed to check their suitability, using the mean values corresponding to tooth height and width plus twice their equivalent standard deviations obtained during the first phase (Figure 2). This was performed following the methodology carried out during the sample selection and the previous phase.

The following study variables were taken into consideration during both phases of this investigation:Anatomical dimensions of the following teeth 55, 54, 53, 63, 64, 65, 75, 74, 73, 83, 84, and 85, as shown in Figure 1. The measurements of the radiographic variables were carried out in duplicate, thus calculating the intraclass correlation coefficient between both measurements to evaluate the intraobserver reproducibility (phase I);Visualization of the coronal portion of the teeth under study and furcation of all primary molars (phase I vs. phase II);Presence of ischemia in the areas close to the radiographic plate (phase I vs. phase II);Presence or absence of gag reflex (phase I vs. phase II);Acceptance/refusal of the radiographic technique by the pediatric patient (phase I vs. phase II).

### 2.3. Statistical Analysis

Descriptive statistics were performed by calculating relative and absolute frequencies for all qualitative variables. Inferential statistics was carried out by calculating the 95% confidence intervals for the proportion of patients with correct visualization of left and right tissue. The same inferential statistics was also performed for the proportion of patients that presented ischemia, gag reflex, and refusal of the radiographic plate. Lastly, a comparison analysis of these proportions between the two study phases (Phase I vs. Phase II) was performed applying the chi-square tests. The significance level was set at 5%. Statistical analyses were performed with STATA version 14.2 (Stata Corp, LLC, College Station, TX, USA).

To determine the intra-observer reliability, a pilot study was carried out prior to the development of this research, re-evaluating 10% of the radiographs 1 week after the first inspection. The average reliability was 96%.

## 3. Results

The total number of patients participating in the study was 80 between the ages of 3 and 5 years old, equally distributed in both phases. The average age corresponding to participants in phase I was 4.22 ± 0.80, while, in phase II, it was 4.12 ± 0.75 years.

### 3.1. Optimal Radiographic Plate Size

Table 1 shows the results obtained for the radiographic dental measurements. The optimal dimensions for the plaque were determined to be 19.5 mm high and 27.3 mm wide.

### 3.2. Comparison of Phase I vs. II

The results obtained regarding the visualization of the dental tissues in both hemiarchs (right and left) are shown in Table 2.

When comparing the presence or absence of soft-tissue ischemia near the edges of the radiographic plate (Table 3), the data indicate that 95% of the patients who bit the size 0+ radiograph showed it. However, the presence of ischemia with the cropped radiograph was 0%. The significance obtained for this variable (phase I vs. II) was *p* < 0.001.

When studying the presence of gag reflex during the radiographic examination (Table 4), the proportion of patients with a gag reflex was significantly higher in children participating in the first phase (*p* < 0.001).

As for the rejection of the radiographic examination (Table 5) reflected in the visual analogue scale, the number of patients who gave it a bad score was higher in the group who bit the larger plate (*p* < 0.001).

## 4. Discussion

The bitewing radiographic technique continues to be the reference test or “gold standard” in pediatric dentistry for the diagnosis of hard-tissue lesions in the oral cavity and the control of the therapy evolution over time. However, aspects such as the partial or superimposed visualization of dental tissues, caused in part by the lack of collaboration of the child during the development of the technique, undoubtedly ruins its diagnostic efficacy. The possibility of triggering the gag reflex or the excessive pressure of the plaque inflicted over the child’s delicate mucous membranes adds to the unfavorable behavior.

Bitewing examination is recommended even in populations with low caries prevalence as a preventive measure. When the risk is high at the age of 4 years, radiographic recordings are indicated [1]. These are small oral cavities and possibly not compatible with the current radiographic size.

Another topic of debate that arises regarding the bitewing technique is the use or not of positioners, always with the aim of improving the quality of the registration and avoiding the disadvantages that derive from it. However, Herman et al. tested two types of film holders, KWIK-BITE (Hawe-Neos Dental, Bioggio, Switzerland) and Snap-a-ray (Dentsply RINN, Charlotte, NC, USA), and compared them with the adhesive-based technique. They found no major differences between them. The most commonly used film holder with small and uncooperative children was the Snap-a-ray type as it decreases the pressure exerted by the film on the oral mucosa and allows adjusting the mesio-distal length of the film to the dental arch length by bending the film, but these radiographic recordings showed a higher tooth overlap. This study showed a high frequency (76.5%) of technical errors regardless of the type of film holder, thus indicating the need for a new medium-size film to help uncooperative children [11]. Pietro et al. suggested the creation of smaller pediatric film holders, and we insisted on the improvement of only reducing the size of the radiographic plate [12].

Ozdemir et al. [13] evaluated children’s pain perception after intraoral radiograph placement in the mouth using the Wong–Baker FACES pain rating scale and the visual analogue scale; however, the data are not comparable to ours as the pediatric population was of a higher age range (6–12 years). Analogue plates, sensors, and phosphor plates were used. The sensor caused more discomfort in the children (*p* < 0.05), and no statistically significant differences were found between the plates; however, these authors indicated that more comfortable devices are required for pediatric patients.

The gag reflex is an innate and involuntary defense mechanism that prevents the entry of foreign bodies via the respiratory tract. In pediatric dentistry, it can get triggered due to general factors such as age, psychological factors such as anxiety and fear (59%) [11,14,15], or anatomical factors such as impression procedures (95.4%) or when foreign objects are introduced into the child’s mouth by touching the lateral edge of the tongue or palate [16]. According to Katsouda et al. [6], 20% of children between the ages of 4 and 12 years present this reflex, which is why it is considered a fairly common obstacle that pediatric dentists encounter, reducing the effectiveness of preventive, diagnostic, and therapeutic procedures and encouraging the child to reject the need for care. Its management is complex [17], and the scientific literature provides a wide range of possibilities: antiemetics, sedatives, local anesthetics, relaxation techniques, distraction, desensitization, acupuncture, acupressure, or the use of low-intensity laser alone or associated with specific acupuncture points [16]. The technique of inverse or extraoral radiography has even been developed [8], doubling the exposure time, but leaving aside the principle of the minimum possible ionizing radiation, since it is necessary to optimize its application to avoid unhealthy exposures [4]. These alternatives are of limited usefulness when our patient is young [7,18], partly because they are applied to older children. According to the results of this study, we believe that the best alternative to avoid this problem when introducing the plate into the oral cavity would be to promote the marketing and use of a trimmed plate. We also agree with other authors [17] that behavioral management such as distraction is another way to control this reflex. We believe that the use of self-affirmation techniques [18], although they may favor the child’s collaboration, do not allow control of the gag reflex, unless it is caused by an emotional factor. Therefore, we insist on the benefit of the smaller radiographic plate.

The cooperation of our patient is essential to establish communication, calm fear and anxiety, and achieve a good radiographic technique and quality dental care [16].

We consider the lack of an increased sampling power our main study limitation. We encourage future research on the basis of the proposed radiographic size.

One of the strengths of this research is the innovation represented by the use of a new plate suitable for the reduced size of the oral cavity present in children with primary dentition. This fact is magnified by the clinical reality it represents and the increasing demand for care in the pediatric population with carious pathology.

## 5. Conclusions

The appropriate dimensions of a radiographic plate allowing the correct visualization of dental tissues and offering an adequate fit in the oral cavity of children between 3 and 5 years of age without causing discomfort or rejection when performing a bitewing radiograph are 19.5 mm in height and 27.3 mm in width.

The trimmed plate is more effective controlling behavioral management during radiographic examination, as it is more accepted by children in the primary dentition because it does not cause ischemia or promote nausea.

## Figures and Tables

**Figure 1 ijerph-19-15413-f001:**
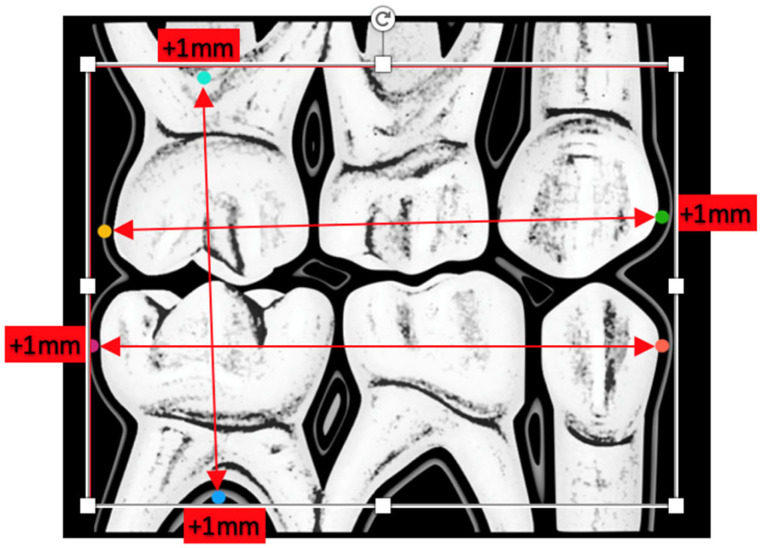
Anatomical points and lines taken as reference for the future size of the radiographic plate.

**Figure 2 ijerph-19-15413-f002:**
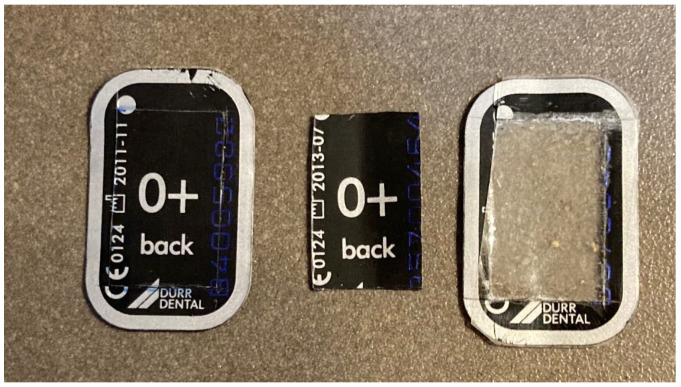
X-ray plate number 0+ untrimmed and trimmed.

**Table 1 ijerph-19-15413-t001:** Radiographic tooth measurements and suitable dimensions of the radiographic plate.

Category	Height	Width
*n*	40	40
Mean	18.16	24.04
Standard deviation (SD)	1.49	1.62
Minimum	13.45	20.75
p25	17.47	23.15
p50	18.37	23.85
p75	18.75	25.07
p90	19.47	26.3
Maximum	22.5	27.6
p normality test	0.0156	0.2907
Optimal dimension ^1^	19.48	27.27

^1^ Mean + 2SD if the variable follows a normal distribution or p90 if the variable does not follow a normal distribution.

**Table 2 ijerph-19-15413-t002:** Comparison of tissue visualization on the left and right sides between phase I and phase II.

Visualization of Dental Tissues on the Left				Visualization of Dental Tissues on the Right			
Total							
Category	*n*	%	IC 95%	Category	*n*	%	IC 95%
No	10	12.5		No	17	21.25	
Yes	70	87.5	80.25–94.75	Yes	63	78.75	69.79–87.71
Total	80	100		Total	80	100	
Phase I							
Category	*n*	%	IC 95%	Category	*n*	%	IC 95%
No	5	12.5		No	10	25	
Yes	35	87.5	77.25–97.75	Yes	30	75	61.58–88.42
Total	40	100		Total	40	100	
Phase II							
Category	*n*	%	IC 95%	Category	*n*	%	IC 95%
No	5	12.5		No	7	17.5	
Yes	35	87.5	77.25–97.75	Yes	33	82.5	70.72–94.28
Total	40	100		Total	40	100	
Chi-square test (*p* = 1)				Chi-square test (*p* = 0.412)			

**Table 3 ijerph-19-15413-t003:** Presence or absence of ischemia between phase I and phase II.

Total			
Category	*n*	%	IC 95%
No	42	52.5	
Yes	38	47.5	36.56–58.44
Total	80	100	
Phase I			
Category	*n*	%	IC 95%
No	2	5	
Yes	38	95	88.25–100
Phase II			
Category	*n*	%	IC 95%
No	40	100	
Yes	0	0	0–0
Chi-square test (*p* < 0.001)			

**Table 4 ijerph-19-15413-t004:** Presence or absence of gag reflex between phase I and phase II.

Total			
Category	*n*	%	IC 95%
No	44	55	
Yes	36	45	34.10–55.90
Total	80	100	
Phase I			
Category	*n*	%	IC 95%
No	4	10	
Yes	36	90	80.70–99.30
Phase II			
Category	*n*	%	IC 95%
No	40	100	
Yes	0	0	0–0
Chi-square test (*p* < 0.001)			

**Table 5 ijerph-19-15413-t005:** Refusal to radiographic examination between phase I and phase II.

Total			
Category	*n*	%	IC 95%
No	42	52.5	
Yes	38	47.5	36.56–58.44
Total	80	100	
Phase I			
Category	*n*	%	IC 95%
No	2	5	
Yes	38	95	88.25–100.00
Phase II			
Category	*n*	%	IC 95%
No	40	100	
Yes	0	0	0–0
Chi-square test (*p* < 0.001)			

## Data Availability

Not applicable.

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
