# Peer review of "Study Regarding the Optimal Dimension of Intraoral Bitewing Radiographs in Patients with Primary Dentition"

_ijerph, 2022, doi:10.3390/ijerph192215413_

Round 1
Reviewer 1 Report
I was pleased to review the article ID ijerph-2043533 entitled “Study regarding the optimal dimension of intraoral bitewing radiographs in patients with primary dentition” for the International Journal of Environmental Research and Public Health. The article investigates the optimum dimensions of a radiographic plate to allow correct visualization of dental tissues and correct fit in the oral cavity of children.
The article is well-written and organized.
Inform in the “2.1. Study population, selection criteria and sample size” how many patients were examined.
Insert figures showing the original and trimmed plate used.
Why did the authors not consider applying phase II in the same population as phase I?
Did you find any difference using the sex of patients as a variable?
Author Response
The authors of this manuscript welcome suggestions, which will undoubtedly improve the quality of the manuscript.

Reviewer 2 Report
Thank you for the opportunity to review. The work in my opinion is well written. I see very minor corrections.
Congrats to the authors of the idea.
L4 - The author Fernando Costa-Ferrer has affiliation number 3 - such a number does not appear below.
L4 - There is an Fernando Costa-Ferrer next to the author † which is not explained below.
L22 – ‘’ statistically. ‘’ – delete the period.
L46 – ‘’… patient.’’ - add direct quote to informacion from The European Academy of Paediatric Dentistry
L59 - add research hypothesis.
L144 – ‘’ STATA IC’’ - is this an error ? I am not aware of such a program.
L245 – ‘’ pathology that is becoming’’ – add the period to the end of sentence.
Author Response

(The authors gave the same response as above.)

Reviewer 3 Report
The reviewer really appreciates the efforts of the authors to conduct this study which has good clinical significance. However, there are several scopes for improving the quality of the manuscript. The reviewer would like to suggest the following revision in the manuscript to make it suitable for publication
The title of the manuscript: (Title study) the author intends to express the type of study? Or it is just a typo error?
In methodology: please add exclusion criteria
Author Response

(The authors gave the same response as above.)

Round 2
Reviewer 1 Report
Thanks for accepting the suggestions.
The manuscript is ready the be accepted.